# Using Convolutional Neural Networks to Map Houses Suitable for Electric Vehicle Home Charging

James Flynn [1,*,†] and Cinzia Giannetti [2,†]

1    Materials and Manufacturing Academy, College of Engineering, Swansea University, Swansea SA1 8EN, UK
2    College of Engineering, Swansea University, Swansea SA1 8EN, UK; C.Giannetti@Swansea.ac.uk
*    Correspondence: 827380@swansea.ac.uk; Tel.: +44-0744-363-7510
†    Current address: College of Engineering, Swansea University Bay Campus, Fabian Way, Crymlyn Burrows, Skewen, Swansea SA1 8EN, UK.

**Abstract:** With Electric Vehicles (EV) emerging as the dominant form of green transport in the UK, it is critical that we better understand existing infrastructures in place to support the uptake of these vehicles. In this multi-disciplinary paper, we demonstrate a novel end-to-end workflow using deep learning to perform automated surveys of urban areas to identify residential properties suitable for EV charging. A unique dataset comprised of open source Google Street View images was used to train and compare three deep neural networks and represents the first attempt to classify residential driveways from streetscape imagery. We demonstrate the full system workflow on two urban areas and achieve accuracies of 87.2% and 89.3% respectively. This proof of concept demonstrates a promising new application of deep learning in the field of remote sensing, geospatial analysis, and urban planning, as well as a major step towards fully autonomous artificially intelligent surveying techniques of the built environment.

**Keywords:** deep learning; electric vehicles; transfer learning; remote sensing; Google Street View

## 1. Introduction

Machine learning and computer vision algorithms have proven to be powerful tools for classification of remotely sensed imagery. Over the past decade, as distributed smart systems and the Internet-of-Things have become increasing widespread, new data analytics technologies have emerged to deal with the vast amount of data produced by these systems. The rapid development of smart devices, artificial neural networks, and computer vision has meant recent approaches to image classification in remote sensing rely increasingly on machine learning techniques [1–4]. Computer vision algorithms have proven to be especially powerful tools due to their versatility, scalability, and low cost [5]. Despite the large amount of research into Land Use and Land Cover (LULC) classification using remote sensing imagery, as well as considerable research efforts into planning smart grid infrastructures for future smart cities [6], the identification and classification of private off-street parking represents a gap in the literature. With Electric Vehicles (EV) emerging as the dominant form of green transport in the UK, it is critical that we address this research gap in order to better understand existing infrastructures in place to support the uptake of these vehicles.

Following the Paris Climate Agreement in November 2016, the UK government made a series of announcements to tackle greenhouse gas emissions. The production and sales of petrol and diesel engine vehicles are to be banned in the UK by 2040, and Clean Air Zones (CAZ) are to be introduced across major cities and local authorities [7,8]. Battery Electric Vehicles (BEVs) and Plug-in Hybrid Electric Vehicles (PHEVs) are exempt to these government restrictions and, depending on $CO_2$ pipeline emissions, plug-in vehicle grants are available for UK customers [9]. These factors, combined with their low running costs

and increasing affordability, make BEVs and PHEVs serious contenders in the future of UK sustainable personal mobility.

The UK government recognize that the availability of accessible and affordable home charging options is key to increasing the uptake of plug-in vehicles. At present, the available evidence suggests that most plug-in vehicle owners will carry out the largest proportion of their charging at home [10,11]. For this reason, the updated 'UK Electric Vehicle Homecharge Scheme' was introduced this year offering a grant of up to £500 to support customers towards the purchase and installation of a home Electric Vehicle (EV) charging point [11]. In order to qualify for the scheme, customers who are the registered keeper, lessee, or have primary access to an electric vehicle must have designated private off-street parking accessible at all times [11].

Even the most extensive mapping agencies in the UK such as Ordnance Survey (OS) and Open Street Maps (OSM) lack accurate data describing detailed features of individual residential buildings. While both datasets do include some building attribute data such as building type, active frontage, square footage, etc., some of this information can be difficult to obtain for various reasons. Firstly, obtaining data relating to street frontage using traditional survey methods is often both costly and time consuming, therefore this information is difficult to keep up to date [12]. Secondly, in the late 1960's following advances in aerial surveying, drawing, and printing techniques, as well as the introduction of digital technologies, OS began to manually digitalize their historical paper maps into what is now the OS Master Map. These techniques, particularly the use of aerial remote sensing, make it difficult to classify buildings or identify their frontal features without ancillary data [13]. This is especially true for identifying areas suitable for EV charging points as it is difficult to determine whether a property has road access from an aerial view, and integral/semi-detached garages may be concealed. Thirdly, mapping agencies such as OSM that predominately rely on Volunteered Geographical Information (VGI) find that the accuracy and attention to detail of various measurements differs between areas depending on the skills and patience of the contributor [14]. Furthermore, the quality of VGI coverage drops considerably in suburban areas where there are fewer contributors [14]. These challenges make it difficult for companies to obtain data on residential areas suitable for EV charging points.

In this paper, we overcome these challenges and present an end-to-end workflow based on deep learning to identify homes suitable for EV charging. Our workflow represents a major advancement in automating the process of surveying the external characteristics of residential properties. This paper also presents the first case of using deep Convolutional Neural Networks (CNN's) to classify residential off-street parking from remotely sourced imagery.

The paper is structured as follows. In Section 2 we review the current literature of machine learning in remote sensing. Following the literature review, Section 3 outlines the methodology to identify residential properties for EV charging. This methodology is presented as a five step, end-to-end workflow with each step described in detail within this section. The methods used to select, train, test and optimise the two CNN's that are utilised within this workflow are also discussed in the methodology. Section 4 includes the results of the CNN optimisation as well as a demonstration of the full workflow on two urban areas. Finally, Sections 5 and 6 include the discussion and conclusion of our findings. A list of acronyms used throughout the document is shown in Table 1.

**Table 1.** A list of acronyms and abbreviations used throughout the paper.

| List of Acronyms | |
| --- | --- |
| LULC | Land Use and Land Cover |
| EV | Electric Vehicle |
| OSM | Open Street Maps |
| OS | Ordnance Survey |
| GSV | Google Street View |
| VGI | Volunteered Geographical Information |
| CNN | Convolutional Neural Network |
| DL | Deep Learning |

## 2. Related Work

### 2.1. Deep Learning in Remote Sensing

Over the past decade, Deep Convolutional Neural Networks (CNN's) have emerged as a powerful tool for remote sensing applications due to their superior performance in visual pattern recognition, object recognition, image classification, and image segmentation [15–17]. Based loosely on the perception mechanisms of the visual cortex, a CNN architecture consists of multiple layers of artificial neurons in a stacked arrangement to perform three main operations: convolution, non-linearity, and pooling/subsampling [15,17]. For image classification, a series of multispectral images are fed into the first layer in the form of 2-D arrays. The following sequential layers are then represented as input and an output feature maps calculated by alternatively stacking convolutional and pooling layers. The final layer is a fully connected layer in which classification is performed. AlexNet, GoogleNet, VGG, and ResNet are the most common pre-trained network architectures in the reviewed literature.

AlexNet is the most commonly used network for remote sensing and LULC tasks due to its high efficiency and performance [18]. AlexNet is an eight layer CNN developed by Krizhevsky et al. in 2007 as part of the ImageNet LSVRC-2010 contest. AlexNet was the first network to incorporate the Rectified Linear Activation Function (ReLU), which marked a major algorithmic change that greatly improved the performance of feed-forward networks and permitted the development of very deep neural networks [19]. The network features five convolutional layers and three max pooling layers that result in approximately 60 million parameters, making it the least complex network reviewed in this section. Due to this relatively low complexity, modified versions of AlexNet have been developed to run on embedded devices in real time [20]. In their 2017 paper, Amato et al. developed a CNN for parking lot occupancy detection based on AlexNet's architecture, which was able to run in real time on a Rasbperry Pi 2 model B [20].

In 2015, the architectural concept of "Inception Modules" was introduced with the development of the GoogleNet network, which won the ILSVRC14 competition [21]. By enabling multiple feature extractors to branch from a single layer, the inception modules allow for filters of different resolutions to operate in parallel. With different sized filters at each layer, more accurate spatial information is retained while the number of parameters is significantly reduced when compared to similarly performing non-inception architectures. This means that despite its 22 layers, GoogleNet has only 4 million parameters, making it less-sensitive to over-fitting. GoogleNet has been demonstrated to outperform other networks in LULC classification using aerial images [22]. A fine-tune transfer learning approach was shown to be the most effective training method when compared to training from scratch or using feature vector [22].

The concept of a Residual Network (ResNet) was introduced in 2016 by He et al. as part of an attempt to overcome degradation when building very deep networks [23]. Degradation refers to the tenancy for training error to suddenly increase as a networks depth is increased beyond the point where accuracy saturates. This issue is overcome by introducing feed-forward shortcut connections around multiple stacked layers within the network architecture. This simplifies the process of mapping the identity function onto the output of the skipped layers, meaning additional layers can be added without increasing

the number of parameters or computational complexity. Residual networks have been demonstrated to outperform other unsupervised methods at measuring urban tree cover from Google Street View (GSV) images [24,25]. Mapping urban green spaces using machine learning techniques has been extensively studied and often use GSV images due to the higher resolution and ground details not found in aerial remote sensing imagery [25–27].

Kang et al. compared a range of CNNs in their ability to classify buildings from GSV and aerial images at several cities in North America [3]. The paper looks at four CNN's: VGG16, AlexNet, ResNet18, and ResNet34. All four networks showed high levels of error when trying to distinguish between houses and garages. The paper identifies the source of this error being due to the high number of images of residential houses with an integral garage. VGG16 performed the best at the classification task followed by the ResNet variants and AlexNet. In this paper, we do not consider VGG networks as they are too computationally expensive.

The pre-trained networks discussed in this section are all trained on ImageNet, a huge dataset containing millions of images and 1000 categories [28]. When re-purposing these pre-trained networks for remote-sensing applications, a common challenge is the lack of publicly available training data [22]. This issue is common across many different fields and solutions have been widely explored in the literature [29–31]. The most common method of adapting pre-trained networks is fine-tune transfer learning whereby the user only adapts a small number of high-level layers to the new task. This means that the network retains knowledge of lower level features that are widely transferable across tasks. An example of this is shown by Castelluccio et al. who demonstrated how a network trained using the fine tune approach outperforms the same network trained from scratch on the same dataset in both final accuracy as well as computation time [22].

### 2.2. Challenges and Limitations of Prior Research

As yet, no attempts have been made to use CNN's to identify residential properties suitable for EV charging from streetscape imagery. Furthermore, there is also a lack of work that has been carried out on applying machine learning techniques to identify external characteristics of individual residential properties. This being said, the challenges of such a task are widely discussed in the literature, and based on similar works we highlight some issues we expect to face in our own research. The first major issue faced by similar works is occlusion, in which features in the target domain are obscured by structures, trees, vehicles, or other objects in the foreground. Proposed solutions include using aerial imagery as auxiliary sources [1,32], or using multiple streetscape images from different angles and positions [5]. Both of these solutions incur additional challenges including increased complexity, labour, and time. The developers of the ImageNet database addressed this issue by classifying images in the target domain regardless of occlusion [28]. The second major problem involves the diversity of the target class. Some structures, such as religious buildings, are easier to classify as they often contain unique, distinctive features that not associated with other classes e.g., spire or a dome. Private driveways vary greatly in size, shape, texture, and their location relative to the associated property, making them difficult to distinguish from other features such as a front garden or pathway. Furthermore, while one would assume that garages could be easily distinguished by a large, metallic, square door, previous attempts to classify garages have proven difficult and are often misclassified as houses as the two often appear in the same image. This issue is highlighted by Kang et al., when four different CNN's demonstrated poor accuracy when attempting to identify garages with a significant portion misclassified as houses [3].

### 3. Methodology

This section introduces our method for remotely surveying urban areas to identify residential properties suitable for EV charging using deep learning. There are no freely available datasets that are suitable to train a CNN to recognize images of residential homes suitable for EV charging point installation. Therefore we first introduce two separate

datasets that have been developed for the purpose of this paper. The first dataset is used to train a CNN to classify GSV images into one of four categories: car parks, trees and foliage, images taken at unsuitable headings, and images of residential properties. The second dataset is used to train a second CNN to distinguish between residential properties that are suitable for EV charging, and those that are not. By splitting up the image classification task into these two discrete problems we are able to use much smaller datasets than if we wanted to train a single CNN to perform the whole task. Further details of how these datasets were developed are discussed throughout this methodology section. The end-to-end methodology, illustrated by the workflow diagram in Figure 1, is split up into five main steps. In steps 1 and 2, a MATLAB code is used to request the geographical and image data for the area to be surveyed. In step 3, this image data are then passed to CNN 1, which performs image classification and discards all images other than those of residential properties. These images are then passed onto CNN 2 in step 3 to perform a second image classification to identify which of the remaining images contain homes suitable for EV charging point installation. Finally, in step 5 a list of all EV suitable homes is presented to the user, as well as a map of their locations. These steps are described in further detail in this section, as well as the methods used to select, train, test, and optimise the two CNN's.

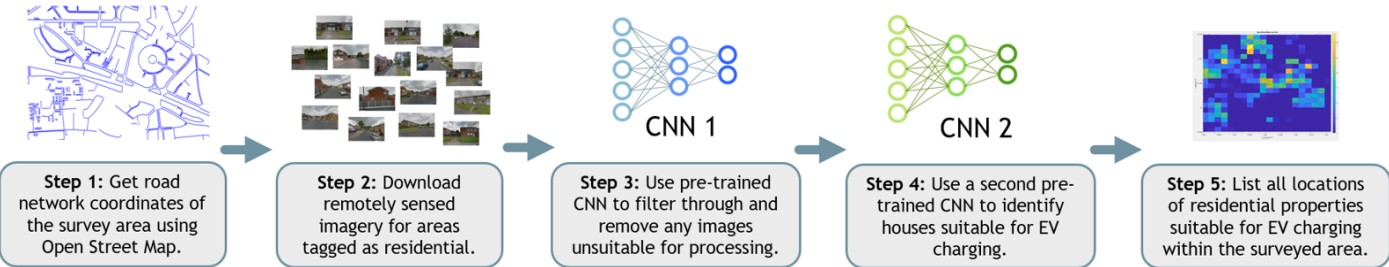

**Figure 1.** Diagram showing the total workflow of the proposed method.

### 3.1. Training and Testing Data Acquisition

The first dataset developed for this paper is used to train and test CNN 1 to identify images of residential properties. The second dataset is used to train and test CNN 2 to distinguish between residential properties that are suitable for EV charging, and those that are not. Both datasets are developed using the practices outlined in this subsection, and consist solely of GSV image data. A breakdown of the number of images in each dataset are shown in Tables 2 and 3. Examples of the image classes used in each dataset are shown in Figure 2. The images used to develop both datasets were collected from seven different UK towns and cities, as shown in Figure 3. Image data were also collected from an additional two cities to be used for further testing of the full workflow, presented as separate case studies. These additional two testing datasets were built using the same practices presented in this section and are discussed in further detail in Section 4. All towns and cities were selected due to their varying local histories, population densities, and geographical locations, which in turn affects the age, density, and style of housing. By gathering as many different instances of images in the target domain, we can build a diverse training dataset despite the small size and therefore capture the diverse range of features found in our target class. OSM provides open source geographical data and was used to acquire the geographic information for these urban areas [33]. At each location, described by rectangular boundaries of longitude and latitude, data are exported in XML format before using the method developed by [34] to extract the road network coordinates. For every coordinate on the road network four separate images were downloaded using the Google Street View API at headings of 0, 90, 180, and 270 degrees, each with an image size of 600 × 400 pixels. The pitch was kept constant at 0 degrees. This same method is used to acquire both the training and testing data, although for the testing data the road network coordinates were interpolated to reduce the distance between nodes and

ensure full coverage of the road network. Testing data were gathered from three locations: Oswestry, Petersfield, and Birmingham. The Oswestry data are used for hyper-parameter optimisation and network selection, while the Petersfield and Birmingham datasets are used as case studies to demonstrate the application of the workflow to survey an urban area for homes suitable for EV charging.

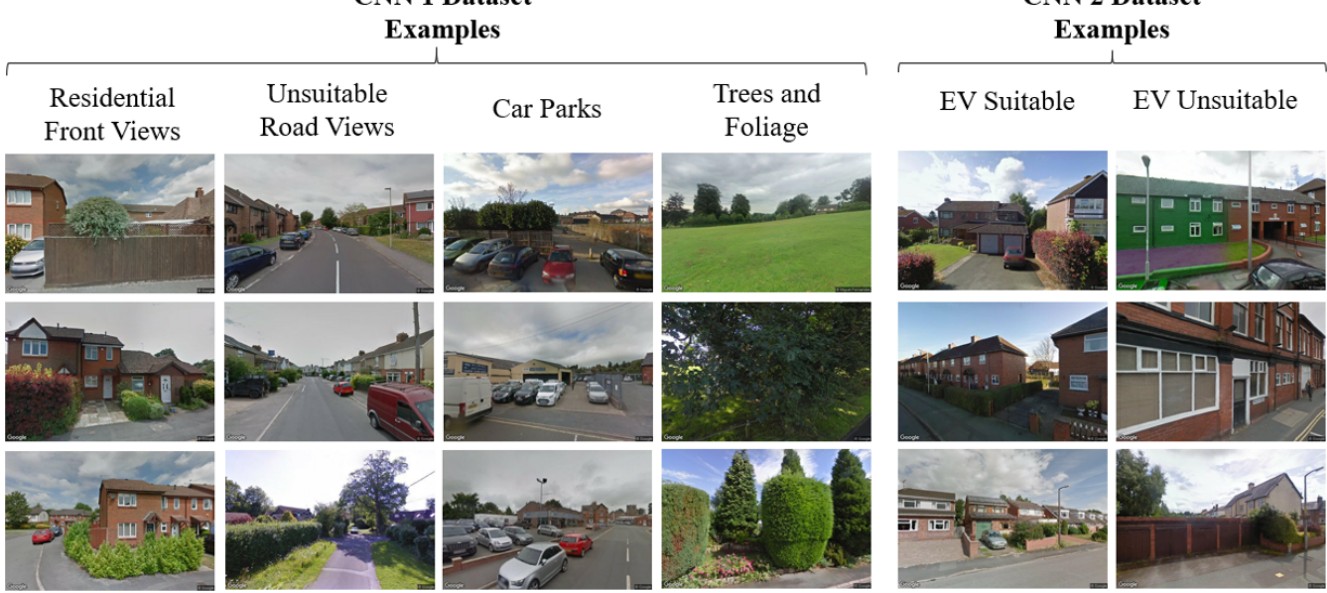

**Figure 2.** Example images from each of the categories in the two datasets.

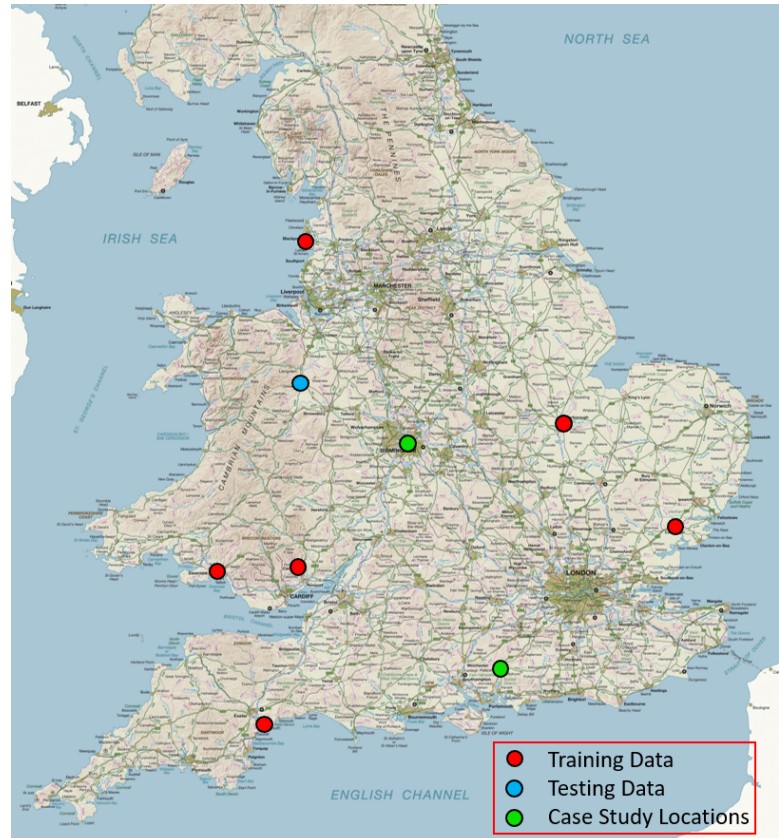

**Figure 3.** A map showing the locations of the towns and cities from which the training and testing data are sourced. This image is adapted from the following source http://ontheworldmap.com/uk/england/map-of-england-and-wales.html (accessed on 8 March 2021).

**Table 2.** A breakdown of the image dataset used to retrain Convolutional Neural Network (CNN) 1. The Oswestry data are used for validation and hyper-parameter optimisation.

| Category | B'pool | P'borough | S'sea | C'bran | Co'ster | Ex'th | Os'try | Totals |
|---|---|---|---|---|---|---|---|---|
| Car Parks | 349 | 424 | 308 | 405 | 90 | 117 | 200 | 1893 |
| Trees and Foliage | 318 | 1658 | 573 | 311 | 348 | 402 | 200 | 3810 |
| Road Views | 354 | 388 | 434 | 334 | 337 | 375 | 200 | 2422 |
| Residential Front View | 1251 | 937 | 947 | 1019 | 972 | 1033 | 200 | 6359 |
| Totals | 2272 | 3407 | 2262 | 2069 | 1747 | 1927 | 800 | 14,484 |

**Table 3.** A breakdown of the image dataset used to retrain CNN 2. The Oswestry data are used for validation and hyper-parameter optimisation.

| Category | B'pool | P'borough | S'sea | C'bran | Co'ster | Ex'th | Os'try | Totals |
|---|---|---|---|---|---|---|---|---|
| EV Suitable | 627 | 743 | 295 | 451 | 898 | 759 | 500 | 4273 |
| EV Unsuitable | 542 | 379 | 411 | 94 | 322 | 415 | 500 | 3232 |
| Totals | 1169 | 1122 | 706 | 545 | 1220 | 1174 | 1000 | 7505 |

Given that the minimal requirements for home EV charging is having access to private off-street parking, we decided that the inclusion criteria for EV suitable properties was an identifiable private driveway or attached/integrated garage [10,11]. To address the issue of object occlusion, one of the major challenges identified in the literature review, it was decided to include all images where target features are occluded by objects in the foreground. Examples of acceptable and unacceptable properties are shown in Figure 2. It is important to note that this image dataset was built by researchers and not authorised home-charging installers. Given that this methodology is intended to be used for a tool to support marketing and urban planning applications, we believe this approach is sufficient to demonstrate the how deep transfer learning can be used for automated remote surveying of external building characteristics.

### 3.2. Network Selection

For both networks, since the training datasets are not large enough to train a network from scratch, a fine-tune transfer learning approach is adopted. For image classification, this approach makes use of existing networks that have already been pre-trained on large image datasets and therefore have already leant certain low-level features such as object edges, shapes, corners, and intensity. Based on related work identified in the literature, it was decided that for both CNN 1 and CNN 2 the performance of three different pre-trained deep CNN's is to be compared: ResNet-18, GoogleNet, and AlexNet. When comparing and optimising network performance, the same two hyper-parameters are optimised using a grid search approach by varying the mini batch size at 32, 64, and 128 while the initial learn rate is varied at $1 \times 10^{-3}$, $1 \times 10^{-4}$, and $1 \times 10^{-5}$. For all runs, the learn rate drop period is set at 15 epochs with a drop factor of 0.1 and the L2 Regularization is set at $1 \times 10^{-5}$ to reduce over-fitting. All other hyper-parameters are set at the MATLAB defaults. Both networks were validated and optimised using the images sourced from Oswestry shown in Tables 2 and 3, which were not used for training. The main metric used to compare each network is overall accuracy, which is defined by the number of correct predictions divided by the total number of predictions. We also consider the recall rate, otherwise known as the true positive rate, which is the percentages of of actual positives correctly identified in each target class. This is particularly important for CNN 1 as we want to maximise the number of images of residential front views that are passed on for processing.

All runs were performed on a single Nvidia V100 PCI 16 Gb GPU. The graph in Figure 4 shows how the three networks compared in their performance following the grid search experiment. For CNN 1, when tested on a sample of the Oswestry dataset containing

200 images per category, GoogleNet achieved the highest overall accuracy of 93.9% when using a mini batch size of 32 and an initial learn rate of $1 \times 10^{-3}$. Overall accuracy is shown in the bottom right hand corner of the confusion matrices in Figure 5. More importantly, this network achieved a recall of 100% at identifying images of Residential Front Views suitable for processing. Recall rate for each class is shown in the bottom row of the confusion matrices. For CNN 2, when tested on a sample of the Oswestry dataset containing 500 images per category, GoogleNet achieved the highest overall accuracy of 93.7% when using the same mini batch size and initial learn rate of 32 and $1 \times 10^{-3}$ respectively. The confusion matrix for the optimised CNN 2 network is shown in Figure 5. Given the results of this experiment, the GoogleNet architecture was selected with a mini batch size of 32 and an initial learn rate of 0.001 to be used for both CNN 1 and CNN 2.

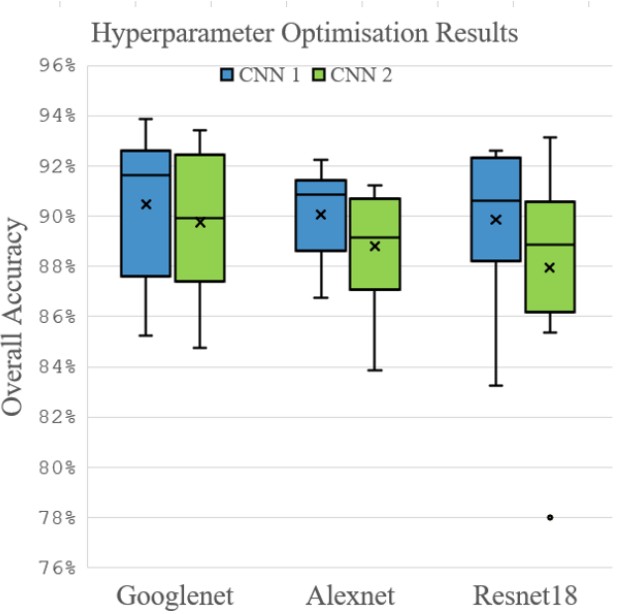

**Figure 4.** Results of the grid search to optimise the hyper-parameters for the three networks.

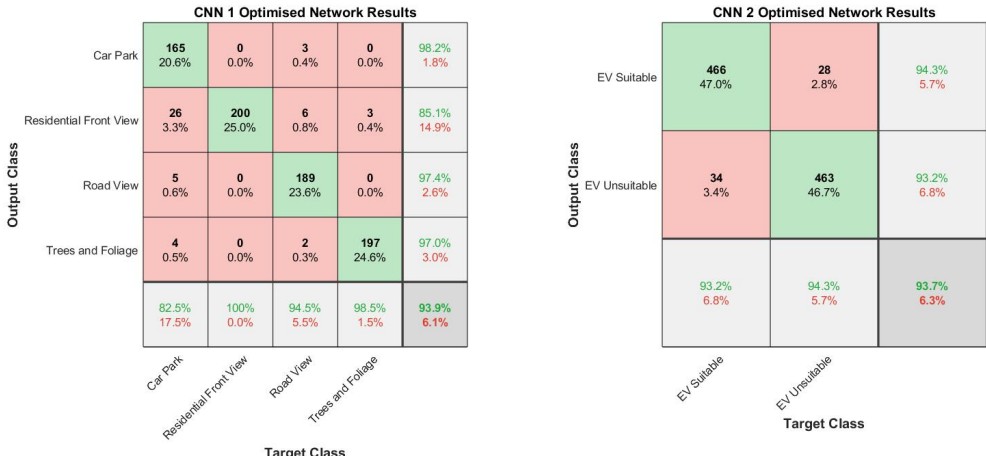

**Figure 5.** Confusion matrices from the best performing networks following the grid search optimisation for CNN 1 (**Left**) and CNN 2 (**Right**).

### 3.3. Data Post-Processing

For testing of the Petersfield and Birmingham datasets, we modified the 'openstreetmap' interface code from [34] to request image data for every 20 m of the road network. This ensures that we downloaded all GSV images of the entire road network.

When each image was downloaded from Open Street Maps, additional properties associated with the requested coordinate were also downloaded and tagged to each image, such as road name and other address data, if available. Once an image was identified by the network as suitable for EV charging, these features were evaluated and returned to the user highlighting all suitable locations within the area, as well as a mailing list of all roads ranked by the frequency of occurrences of EV suitable properties.

## 4. Testing the Full Workflow

In order to demonstrate the entire workflow on a larger scale we were required to manually label every image in each survey area order to evaluate the performance of the system. To make this feasible, all of the images within the test areas were split into three categories: 'EV Suitable', 'Unsuitable (Residential Front View)', and 'Unsuitable (Other)'. These three categories allow us to test the performance of both networks. CNN 1 should filter out all images in the 'Unsuitable (Other)' category, which includes any images of car parks, trees and foliage, and unsuitable headings. As before, images of buildings are then passed onto CNN 2, which analyses the remaining images to identify those suitable for EV charging point installation before the results are then plotted on a the map.

### 4.1. Test Area 1—Petersfield

#### 4.1.1. Data Acquisition

Petersfield is a rural town in Hampshire, South England with a population of approximately 15,000 people according to the most recent census data [35]. After downloading images at each of the four headings and removing any duplicate images, this gives a total of 6433 images in the Petersfield testing dataset. These images were then manually labelled into the three test categories also shown in Figure 6. Image data for the Petersfield test area were requested at 30 m intervals along all roads within the boundary that were tagged as 'Residential', resulting in a total of 2092 locations along the road network shown on Figure 6.

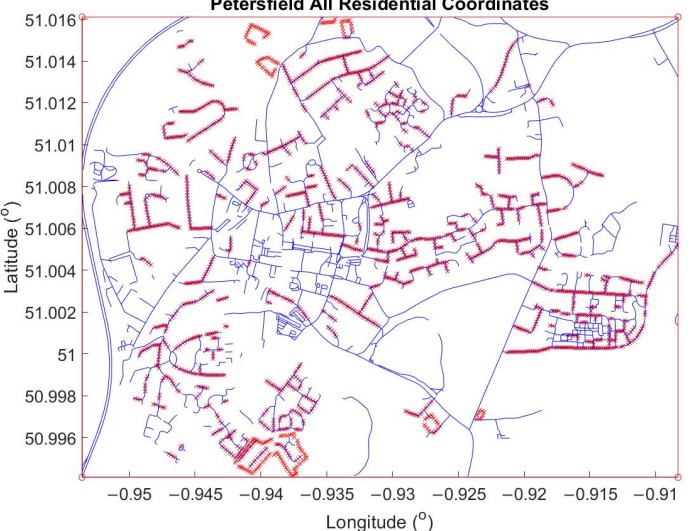

| Category | Petersfield |
|---|---|
| EV Suitable | 1651 |
| Unsuitable (Residential Front View) | 3478 |
| Unsuitable (Other) | 1304 |
| Total | 6433 |

**Figure 6.** The locations of all downloaded images from the Petersfield test area (**Left**) and a breakdown of the image categories (**Right**).

#### 4.1.2. Results

For the Petersfield test area, CNN 1 achieved a recall of 90.7% for identifying images of 'Residential Front Views' suitable for processing, as shown in Figure 7. Following the CNN 1 step, 2851 images were passed onto CNN 2 for processing. CNN 2 then identified

1693 images as suitable for EV charging, achieving a recall of 88.6%, as shown in Figure 7. Overall, the system recognises 1355 of the total 1651 images suitable for EV charging. Within the Petersfield test area, 132 streets were identified with properties suitable for EV charging. Figure 8 shows all suitable properties plotted on a map of the road network. Figure 8 shows the roads returned by the system as having the highest number of residential properties suitable for EV charging point installation.

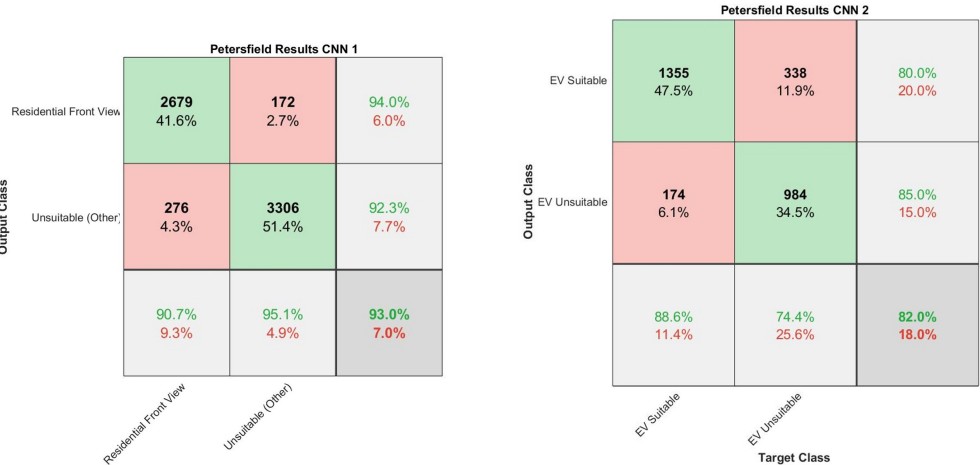

**Figure 7.** The resultant confusion matrices for CNN 1 and CNN 2 on the Petersfield testing data.

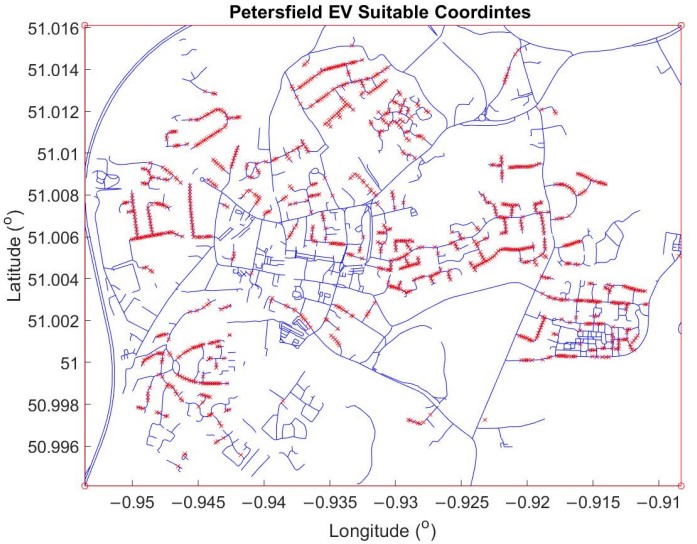

| Road Address | Frequency |
| --- | --- |
| Woodbury Avenue, Petersfield, GU32 | 70 |
| Stafford Road, Petersfield, GU32 | 58 |
| Monks Orchard, Petersfield, GU32 | 56 |
| Hanger Way, Petersfield, GU31 | 48 |
| Tilmore Gardens, Petersfield, GU32 | 45 |
| Marden Way, Petersfield, GU31 | 37 |
| Rival Moor Road, Petersfield, GU31 | 36 |
| Durford Road, Petersfield, GU31 | 36 |
| Cranford Road, Petersfield, GU32 | 33 |
| Moggs Mead, Petersfield, GU31 | 33 |

**Figure 8.** A plot of all coordinates within the Petersfield test area identified as suitable for EV charging (**Left**). A list of the top 10 roads most suitable roads for EV charging ranked by frequency of occurrences (**Right**).

### 4.2. Test Area 2

#### 4.2.1. Data Acquisition

For the second test area, we collected GSV data from a residential area roughly 4 km east of Birmingham city centre. A total of 28,728 coordinates along the road network were downloaded, which resulted in 14,766 in this dataset after removing duplicate images. These images were then manually labelled into the same three test categories as before, as shown in Figure 9. The road network for this test area is shown in Figure 9.

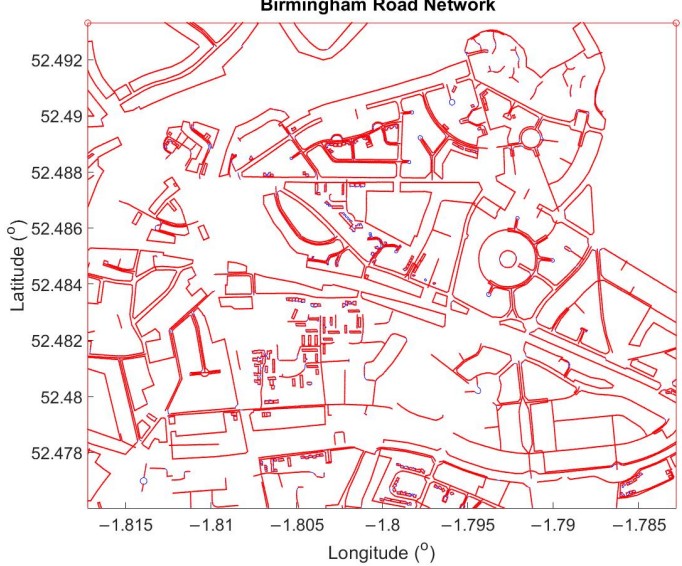

| Category | Birmingham |
|---|---|
| EV Suitable | 5306 |
| Unsuitable (Residential Front View) | 7538 |
| Unsuitable (Other) | 1922 |
| Total | 14,766 |

**Figure 9.** The locations of all downloaded images from the Birmingham test area (**Left**) and a breakdown of the image categories (**Right**).

#### 4.2.2. Results

Once the images were downloaded, they were passed through CNN 1 for pre-processing. For the Birmingham test area, CNN 1 achieves a recall of 96.3% when identifying images of Residential Front Views, as shown in Figure 10. Following the CNN 1 step, 7408 images were passed onto CNN 2 for processing, which then identified 5376 images as suitable for EV charging, giving a recall of 89.3%, as shown in Figure 10. Overall, the system recognises 4617 of the total 5306 images suitable for EV charging. Figure 11 shows all EV suitable properties plotted on a map of the road network as well as the roads identified by the system as having the highest number of residential properties suitable for EV charging point installation.

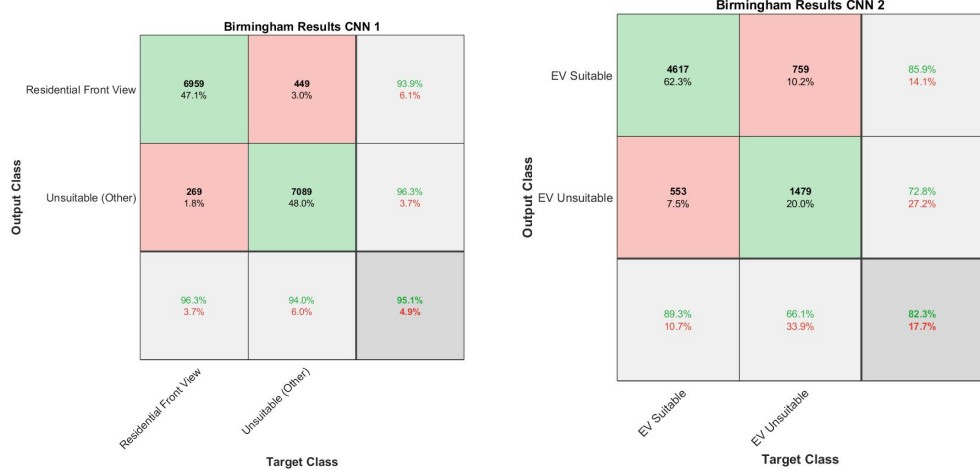

**Figure 10.** The resultant confusion matrices for CNN 1 and CNN 2 on the Birmingham testing data.

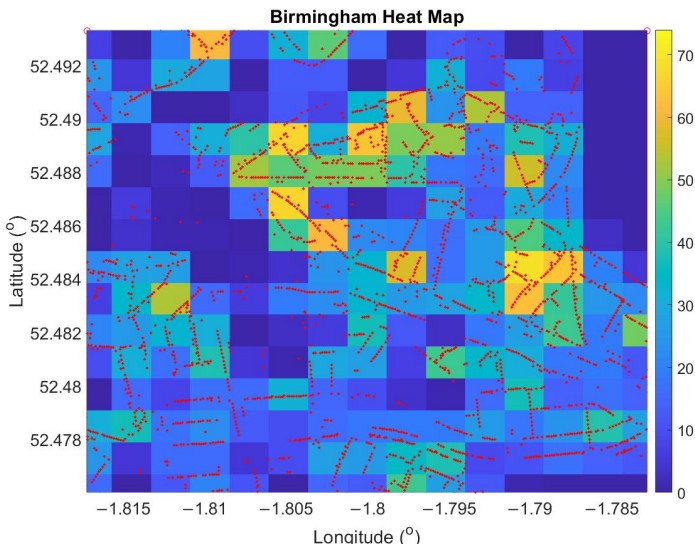

**Figure 11.** A heat map showing the areas with the highest density of Electric Vehicles (EV) suitable properties in the Birmingham test area as well as the individual locations of EV suitable properties plotted in red (**Left**). A list of the top 10 roads most suitable roads for EV charging ranked by frequency of occurrences (**Right**).

| Road Address | Frequency |
|---|---|
| Stud Lane, Stechford, B33 | 44 |
| Peplow Road, Stechford, B33 | 39 |
| Manor Gardens, Yardley, B33 | 23 |
| Kirton Grove, Stechford, B33 | 14 |
| Tanfield Road, Stechford, B33 | 14 |
| Shelly Croft, Stechford, B33 | 14 |
| Croxton Grove, Stechford, B33 | 10 |
| Kington Way, Yardley, B33 | 9 |
| Beswick Grove, Stechford, B33 | 9 |
| Alcombe Grove, Yardley, B33 | 8 |

## 5. Discussion

The system is shown to be highly accurate at identifying EV suitable properties in both the Petersfield and Birmingham test areas achieving a recall of 87.2% and 89.3%, respectively. One of the issues with this current method is that a significant proportion of the original downloaded images are unsuitable for processing. This is partly due to the GSV headings being unreliable, and we must therefore download all four angles to ensure a suitable image of the adjacent property is retrieved. In the Petersfield example, despite only downloading images from areas tagged as residential, only 26% of the images contained properties suitable for EV charging. We are currently not charged for the small number of images we are downloading (<24,000 per month), however, when this method is scaled up to survey local authority areas, Google API applies a charge of approximately £0.0007 per image. Another challenge of this method is that OSM relies on crowd sourced data to develop and update maps. Despite quality assurance tools being in place, mistakes are not uncommon and information such as road names are often missing. Approximately, 94% of the road names are unable to be retrieved using our current method. This explains the low /hlnumber of suitable houses identified for each road as shown in Figures 8 and 11. Despite these challenges, our model proves to be robust when tested on image data sourced from new locations. Furthermore, our model improves on the only previous attempt to identify off-street parking from remotely sensed images, achieving both higher recall and precision rates [3]. It is important to note that in the paper by Kang et al. the target classification class only included images of 'Garages', whereas we include both garages and driveways in out target class.

## 6. Conclusions

We have demonstrated that surveying external building characteristics of small to medium urban areas is possible using open source data. A system was developed to identifying and map residential properties suitable for EV charging with high levels of accuracy. To achieve this, we introduce a novel end-to-end workflow that utilises Open Street Map and Google Street View data, which are processed by two separate CNN's. To select an appropriate architecture for these tasks, the performance of three pre-trained networks was compared by means of a grid search experiment. GoogleNet outperformed AlexNet and ResNet18 at both image classification tasks achieving accuracies of over 90%.

To validate the performance of the entire workflow, we remotely survey two contrasting urban areas to identify all properties suitable for EV charging. By integrating open source geographical data, we are also able to identify which roads have the highest number of suitable properties. In these surveys, our methods improve on previous attempts to classify garages from GSV images, achieving accuracies of around 90%. This paper also presents the first case of using deep CNN's to classify residential driveways from remotely sensed imagery. To provide more detailed address data, as well as improving the efficiency of data acquisition, it is suggested that future works utilise OS data and GIS software when applying these methods at a larger scale. An alternative solution could be to use aerial images as an auxiliary dataset. Despite these limitations, the methods developed in this paper represent a major advancement towards fully automated remote surveying capability to audit the built environment.

**Author Contributions:** J.F. conceived of the presented idea and C.G. verified the analytical methods. J.F. performed the computations and C.G. supervised and validated the findings of this work. Writing, editing, and formatting the manuscript was carried out by J.F. with support from C.G. Funding acquisition was carried out by C.G. All authors have read and agreed to the published version of the manuscript.

**Funding:** This research was funded by European Social Fund via the Welsh Government (c80816)" and and "the Engineering and Physical Sciences Research Council (Grant Ref: EP/L015099/1".

**Data Availability Statement:** Restrictions apply to the availability of these data. Data was obtained from Google using the Google Street View API and are available from the Google Maps Platform at (https://developers.google.com/maps/documentation/javascript/streetview accessed on 12 March 2021).

**Acknowledgments:** Cinzia Giannetti acknowledges the support of the UK Engineering and Physical Sciences Research Council (EPSRC) project EP/S001387/1. The authors would also like to acknowledge the M2A funding from the European Social Fund via the Welsh Government (c80816), the Engineering and Physical Sciences Research Council (Grant Ref: EP/L015099/1), and Ford Motor Company that has made this research possible. We further acknowledge the support of the Supercomputing Wales project (c80898 and c80900), which is partly funded by the European Regional Development Fund (ERDF) via the Welsh Government.

**Conflicts of Interest:** The funders had no role in the design of the study; in the collection, analyses, or interpretation of data; in the writing of the manuscript, or in the decision to publish the results.

## Abbreviations

The following abbreviations are used in this manuscript:

| | |
|---|---|
| MDPI | Multidisciplinary Digital Publishing Institute |
| DOAJ | Directory of open access journals |
| TLA | Three letter acronym |
| LD | linear dichroism |

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
