# Peer review of "Using Convolutional Neural Networks to Map Houses Suitable for Electric Vehicle Home Charging"

_ai, doi:10.3390/ai2010009_

Round 1
Reviewer 1 Report
The paper is well-presented overall. It does present an emerging technology area.
The main concerns suggested are:
- The paper seems to be just implementing or applying Deep learning to Electric vehicles, without making a real substantive innovative contribution.
- What is the scientific contribution of this paper?
- How do the results achieved compare with other related results?
Author Response
"Please see the attachment."

Reviewer 2 Report
This paper is well organised and proposing a solution to the goal of identifying and mapping residential properties suitable for EV charging. I believe that the authors are working on a promising application that could help researchers in the field of autonomous surveying techniques.
I would suggest that the authors double-check the footer of the paper where it is stated that this paper has been "Submitted to Remote Sens. www.mdpi.com/journal/remotesensing" while I believe I am reviewing for AI.
Before the abstract (line 0): Version December 10, 2020 submitted to Remote Sens.
And again, this is repeated at the end of the paper at line 416: © 2020 by the authors. Submitted to Remote Sens.
Please consider the following text corrections:
- line 13: Start section numbering with 1
- line 93 and line 99: Starting a sentence with a numbered citation does not feel right. Please insert the author(s) name before the citation. Please check this around the paper.
- You have to add a space before the citation brackets. e.g. in line 113 add a space between imagery and [24–26].
- Tables 1 and 2 need reformatting (Right margin)
- Figure 5: increase font size.
Author Response
"Please see the attachment."

Reviewer 3 Report
1. "Deep learing" should be added into the part of "Keywords"; 2. Maybe Python can resolve some problems of Matlab in your future works; 3. Some contents in "Discussion" should be moved to the last part of "Conclusion" as "future works".Author Response
"Please see the attachment."

Reviewer 4 Report
Comments to Authors,
The authors did a job to identify residential properties suitable for EV charging based on deep learning. Even though the experiment showed a higher accuracy, the manuscript includes insufficient information and requires a major revision before publication. For the manuscript, my comments are as follows:
- Multiple datasets are mentioned in the paper. What is the relationship between these datasets? There is no clear explanation in the paper. For example, where was the dataset used in step 3 come from?
- How to match the additional properties of the dataset with the adaptability of the charging device? It is not clearly stated in this paper. For example, how to label the image dataset used in step 4.
- How to use the images of the dataset during the screening process of CNN1?
- The positions of the tables and figures are a bit strange, which are far away from the description. In addition, the format is not good.
- Please refer the following papers for possible improvement: 1)” Bilevel Programming Approach for Optimal Planning Design of EV Charging Station” in IEEE Transactions on Industry Applications; 2) "Visual Perception Enabled Industry Intelligence: State of the Art, Challenges and Prospects," in IEEE Transactions on Industrial Informatics; 3)” Urban Traffic Control in Software Defined Internet of Things via a Multi-Agent Deep Reinforcement Learning Approach” in IEEE Transactions on Intelligent Transportation Systems
- The experimental results did not reflect the results of other experimental settings. For example, Resnet-18 and Alexnet.
7.The details need to be checked carefully. For example, there should be no space after the word “coordinates” in line 166.
- More experiments are needed to support the conclusion that the proposed methods are better.
Author Response
"Please see the attachment."

Reviewer 5 Report
- At Line 158, it said GSV data were collected from eight town and cities, which is not agree with points shown at Figure 2 that has nine points. Please check and revise that.
- At Line 188, it said in section 3.1 …,. Based on the manuscript, it should be section 2.1. Please check and revise that.
- It is suggested that the manuscript add some descriptions of the key requirements of be an EV charging point in article. Meanwhile, if possible, one example image of the EV charging point can also be added along with these descriptions. It also suggested that the manuscript describes the criteria or key image features of the EV charging point to make it be differentiated from other images, such as car parks, tree and foliage, road view and residential front view.
- In this study, label of images for the EV suitable point is done by researchers instead of by authorized installers. Please discuss the legality and the practical use of the research findings.
- In manuscript, three forms of term ‘googlenet’ are used and they are googlenet, GoogleNet and GoogLeNet. Please check and revise, if necessary.
- In Figure 10, ‘…EV suitable properties plotted in red’ is used. In Figure 7, may add these similar words to figure title. Meanwhile, in Figure 9, the color for the EV suitable properties may consider to red color so as to agree with color in Figure 7 and Figure 10.
- In Figure 9, the table at right column might miss a Total (Please refer Figure 6). Please check and revise that.
Author Response
"Please see the attachment."

Reviewer 6 Report
This paper propose a deep learning based method to recognize suitable residential properties suitable for Electric Vehicles (EV) charging from street view images. A large dataset from 9 different locations around the entire England was collected by OSM and GSV for model training and testing. To tackle the raw dataset with a great amount of unsuitable images for processing, a cascade model with two deep CNNs is proposed, where the first CNN is used to select images taken at suitable headings and the second CNN is then used to identify all properties for EV charging. A grid search on some hyper-parameters including batch size, initial learning rate, and model type is conducted. The model is trained on data from 6 places and evaluated on 2 contrasting places, Petersfield and Birmingham. Results show that the proposed model can achieve accuracies of around 90%.
Strength:
- The research of this paper is very meaningful, as Electric Vehicles will become more and more dominant and necessary in future transportation.
- A large and diversity dataset from all around the England is collected for model evaluation.
- The accuracies from two places could reach about 90%, which are soundly good.
Weakness:
- I think the mean weakness is that the result is not convincing enough to me, because only two of nine places are evaluated. I suggest the authors conducting a cross validation. For example, each one or two of the nine places is treated as test set and the rest are training set and then report the average accuracy.
- Ablation study should be conducted that one CNN model is directly trained to identify the suitable properties EV charging to demonstrate the effectiveness of the cascade of two CNN model.
- Visualization study on the CNN kernels or Class Activation Mapping (CAM) to explain how and why the CNN model works on the identification of EV charging places would enrich the study.
- In Section 1.2, authors list two limitations of deep learning on the EV charging task including occlusion and diversity of class. How does the proposed method handle these limitations?
- How is the problem of missing information and details tackled in this paper, which is stated in the first paragraph of Section 2.2?
In a word, I think the idea of this paper is good and the performance is promising. However, more experiments and explanations are required to make the paper more clear and convincing. I suggest a major revision.
Author Response
"Please see the attachment."

Round 2
Reviewer 1 Report
The reviews are revised accordingly, and the paper has made some novelty.
Reviewer 4 Report
Comments to Authors,
The authors did a job to identify residential properties suitable for EV charging which based on deep learning. Even though the experiment showed a higher accuracy, the manuscript includes insufficient information and requires a major revision before publication.
For the manuscript, my comments are as follows:
- The position of the table introducing proper nouns is very strange. It is recommended to put it at the beginning of the second page, and the centering will be more standardized.The format also needs to be corrected.
- Adding an image example is a good way of expression, but why is fig.3 behind fig.4?There are inconsistencies and I hope they can be adjusted.
- The experiment lacks a description of the criterion of the experimental results, such as : the definition of “accuracy”, the definition of "frequency”, the difference between the two. In addition, the description should be added to the experimental settings.
- This paper did a job to identify residential properties suitable for EV charging. The paper also mentions lane detection, which all belong to the application of intelligent transportation. Please refer the following papers for possible improvement: 1)” Bilevel Programming Approach for Optimal Planning Design of EV Charging Station” in IEEE Transactions on Industry Applications; 2) "Urban Traffic Control in Software Defined Internet of Things via a Multi-Agent Deep Reinforcement Learning Approach," in IEEE Transactions on Intelligent Transportation Systems。
- Pleasepay attention to not just use text to describe the accuracy rate, but also indicate it in the image. For example: "95.1%" in line 281 should be marked as the recall rate in figure 9. The same problem exists in other parts. Although the article is clear on the whole, I still hope you pay attention to the clarity of the article. There are numerous places in the article that say the same thing over and over again. I hope you can check and delete the redundant parts.
